# An Interdisciplinary Design of an Interactive Cultural Heritage Visit for In-Situ, Mixed Reality and Affective Experiences

**Xabier Olaz** **, Ricardo Garcia, Amalia Ortiz *, Sebastián Marichal** **, Jesús Villadangos, Oscar Ardaiz and Asier Marzo** 

UpnaLab, ISC—Institute of Smart Cities, Public University of Navarre, 31006 Pamplona, Spain;
xabier.olaz@unavarra.es (X.O.); ricardo.garcia@unavarra.es (R.G.); sebastian.marichal@unavarra.es (S.M.);
jesus.villadangos@unavarra.es (J.V.); oscar.ardaiz@unavarra.es (O.A.); asier.marzo@unavarra.es (A.M.)
* Correspondence: amalia.ortiz@unavarra.es

**Abstract:** Interactive technologies, such as mixed-reality and natural interactions with avatars, can enhance cultural heritage and the experience of visiting a museum. In this paper, we present the design rationale of an interactive experience for a cultural heritage place in the church of Roncesvalles at the beginning of Camino de Santiago. We followed a participatory design with a multidisciplinary team which resulted in the design of a spatial augmented reality system that employs 3D projection mapping and a conversational agent acting as the storyteller. Multiple features were identified as desirable for an interactive experience: interdisciplinary design team; in-situ; mixed reality; interactive digital storytelling; avatar; tangible objects; gestures; emotions and groups. The findings from a workshop are presented for guiding other interactive cultural heritage experiences.

**Keywords:** interaction; interdisciplinary design; interactive digital storytelling; emotional conversational avatars

## 1. Introduction

Our aim is to design an interactive tourism experience for The Collegiate Church of Roncesvalles, which is an emblematic spot of the Spanish "Camino de Santiago". Many tourists and experts in history and cultural heritage (CH) visit the place because of its rich past, full of exciting stories.

This audience heterogeneity makes it difficult to efficiently deliver the content [1]. Interactive technologies can help to contextualize real-life scenarios and the content and therefore to create deep connections with the site [2]. Thus, we decided to develop an interactive system able to engage with the users, adapt content and personalize the experience. At the same time, the project has to respect the aesthetics of the space. The objective of this project is to gather design guidelines for an in-situ interactive experience for a cultural heritage place that emotionally engages with the visitors.

To inform the system design we followed a mixed strategy balancing the expert judgment from an interdisciplinary team and the outcome of a participatory design process. The whole process consisted of two brainstorming sessions and one workshop where user experience (UX) techniques, such as the definition of Persona, Scenarios, User Journeys and role-playing, were applied.

As a result of this process, we designed a prototype where the primary interaction is with an avatar that incorporates storytelling techniques and emotional interaction and narrates local stories associated with historical objects placed around the room. Storytelling is driven through the interaction with real objects and with an emotional conversational avatar. Rather than using VR/AR glasses, the avatar (who represents a miner) is projected on a real wall of the church; the wall simulates a mine using 3D mapping projection. In this way, we take advantage of the existing building and provide a device-less experience, enabling group interaction and collaboration.

In this paper, we present the design process by analyzing the output of each brainstorming and workshop session. We discuss the design rationale of the system and contribute with design guidelines for mixed-reality systems in the CH context.

## 2. Related Work

Incorporating technology in museums and cultural heritage (CH) can improve engagement [3–5], user experience [4,6], presence [7] and cultural awareness [3,4].

We have extracted from previous research the elements that are relevant for designing in-situ immersive experiences where the visitors can move around the environment. These elements are shown in Table 1 and are as follows:

- Interdisciplinary Design Team (IDT): multiple types of expertise are needed to design an emotional and immersive experience for the visitor. We analyze the kind of contribution made by each expert;
- InSitu: we looked for previous research with experiences designed to take place in the real CH spot;
- Mixed Reality (MR): using MR technology to augment the CH real place with multimedia content;
- Interactive digital storytelling (IDS): to generate the narrative guided by the visitors interactions;
- Avatar: a humanoid avatar as a tourist guide;
- Objects: using interactive objects, virtual or real, that are around the environment;
- Gestures: follow the user's position and track their corporal gestures to serve as another input for the interactive system;
- Emotions: taking emotion into account during the design process or introducing affective computing technology;
- Groups: managing the interaction of multiple visitors that come as a group.

**Table 1.** Features of Interactive Cultural Heritage found in existing research.

| | IDT | In Situ | MR | IDS | Avatar | Objects | Gesture | Emotion | Groups |
|---|---|---|---|---|---|---|---|---|---|
| Ciolfi & McLoughlin (2012) [2] | ✓ | ✗ | ✗ | ✗ | ✓ | ✓ | ✗ | ✗ | ✗ |
| Flynn (2013) [8] | ✗ | ✗ | ✓ | ✗ | ✗ | ✗ | ✓ | ✗ | ✓ |
| Pujol et al., (2013) [9] | ✓ | ✗ | ✓ | ✓ | ✗ | ✗ | ✗ | ✗ | ✗ |
| Niculescu et al., (2014) [10] | ✗ | ✗ | ✗ | ✗ | ✓ | ✗ | ✗ | ✗ | ✗ |
| Pietroni & Adami (2014) [11] | ✗ | ✗ | ✓ | ✓ | ✗ | ✗ | ✓ | ✓ | ✗ |
| Seppala et al., (2016) [12] | ✗ | ✓ | ✓ | ✓ | ✓ | ✗ | ✓ | ✗ | ✗ |
| Perry et al., (2017) [13] | ✓ | ✗ | ✗ | ✗ | ✓ | ✓ | ✗ | ✓ | ✓ |
| Vosinakis (2017) [14] | ✗ | ✗ | ✗ | ✗ | ✓ | ✗ | ✗ | ✗ | ✗ |
| Nisi et al., (2018) [3] | ✓ | ✓ | ✓ | ✓ | ✓ | ✗ | ✗ | ✗ | ✗ |
| Carrozzino et al., (2018) [4] | ✗ | ✗ | ✗ | ✗ | ✓ | ✗ | ✗ | ✗ | ✗ |
| Schaper et al., (2018) [5] | ✓ | ✓ | ✓ | ✗ | ✗ | ✗ | ✗ | ✗ | ✓ |
| Papagiannakis et al., (2018) [15] | ✗ | ✓ | ✓ | ✗ | ✓ | ✓ | ✗ | ✗ | ✗ |
| Rizvic et al., (2019) [16] | ✓ | ✗ | ✗ | ✓ | ✗ | ✗ | ✗ | ✗ | ✗ |
| Sylaiou et al., (2020) [7] | ✗ | ✗ | ✓ | ✓ | ✓ | ✗ | ✗ | ✓ | ✗ |
| Dionisio (2021) [17] | ✓ | ✓ | ✗ | ✗ | ✗ | ✗ | ✗ | ✗ | ✗ |
| Hammady et al., (2021) [6] | ✗ | ✗ | ✓ | ✓ | ✓ | ✗ | ✓ | ✗ | ✗ |
| Teston & Munoz (2021) [18] | ✗ | ✗ | ✓ | ✗ | ✓ | ✓ | ✗ | ✗ | ✗ |

## 2.1. Mixed-Reality in CH Spaces

The reality–virtuality continuum [19] classifies mixed-reality as the domain between real and virtual environments. Augmented reality (AR) is the part of MR that is closer to real environments, adding information (e.g., visual or auditory) on top of existing real elements.

A common approach to enhance CH experiences is using AR solutions based on mobile devices [3,20–22] using the technique of Window-on-the-World (WoW) where digital images are overlaid on top of the camera feed of the phone [23]. This approach allows visitors to physically explore the CH site while digital content augments the experience when they see through the device. However, this encourages looking at the phone rather than at the physical space, and focuses on an individual experience [24], constraining the possibilities for group collaboration.

Tangible elements placed in the curated space can mitigate the problem of looking at the phone; for instance, barcodes [25], QR codes [2] or visual markers [3]. This makes visitors pay attention to the physical space before accessing the digital content [3]. Another approach to achieving mobile AR was proposed by [26,27]; they used a see-through head-mounted-display (HMD) and mobile computing equipment to deliver a realistic and interactive immersive experience to the visitors. However, to some extent, this kind of mixed-reality technology also hinders group interaction [25].

Some studies aim at removing the need to carry or wear devices so they propose to project over the physical place [5,11]. Projected augmented reality (AR) offers ways of interacting with an exhibition [28] and improving group experience [29]. For these projection based systems, combining mixed reality with storytelling and gamification is important to achieve user immersion [15].

## 2.2. Interactive Digital Storytelling (IDS)

Using IDS generates a multifaceted experience that encourages conversation and interaction with the visitors [30] by changing from a linear narrative to a more dynamic narrative that, for example, enables the adjustment of the detail and the depth of the presented information without affecting the main educational purpose.

Users appreciated interactive storytelling, empathized with the narrator's character, and learned information about a cultural monument in an attractive and immersive way [16]. In their work, a website showed elements of a fortress overlooking the city of Sarajevo with an IDS system.

Yasmine's Adventures [3] mixes storytelling with a physical walk around the area to foster curiosity and willingness to explore the neighborhood [3]. The experience was oriented to individual participants.

For the Etruscanning EU project [11], storytelling and interactive environments are important resources to reconstruct and communicate a cultural context. Their work highlights the importance of interactive storytelling, the definition of adapted content for different audiences and new interaction paradigms to make digital content accessible to all audiences.

CHESS [9] applies content personalization and adaptability to the digital applications of the Acropolis Museum in Athens. This emphasizes the on-site presentation of museum objects through interactive stories personalized to each individual or group of visitors.

Fragments of Laura [3] employs transmedia storytelling to involve tourists in the development of the understanding of cultural heritage. In this case, it uses mobile devices with interactive 360° VR, 2D Motion Comics and Audio Gossips to generate a multimedia story with local awareness and a hypermedia platform. On the other hand, ref. [4] presents three narration alternatives: text panels, audio guides and a virtual guide; the latter improves attention and participation and contributes to better delivery of content and learning.

The use of avatars for the design of an interactive digital narrative [7] gives importance to the social presence, the interrelation between avatars and users and the emotions evoked through different virtual characters that narrate the same story. Thus, the affective potential,

persuasiveness and emotional impact of avatars influence the degree of acceptance of an emotionally charged narrative.

### 2.3. Avatars: Expressive and Realistic Virtual Humans

For an immersive user experience, using a virtual avatar that narrates local stories may guide and make the story more believable to the visitors [7]. Furthermore, a virtual avatar has the data to build on additional historical facts, plus the ability to represent scenes virtually, which helps visitors to understand some concepts and make them more immersive and entertaining [6].

For immersion, having a realistic avatar that resembles a human makes the interaction more believable and relatable for the visitor [31]. These realistic representations make users rate the experience as fairly similar to face-to-face interactions [32]. Avatars can also convey emotions making the interaction more believable [33], and can engage visitors [13].

However, approaching a human-like virtual replica implies some risk, as users may face the uncanny valley [34] when interacting with the avatar. High fidelity characters ended up being considered more attractive, less disturbing, more human and more cheerful than their low fidelity counterparts [35].

Additionally, the real-time capabilities of the avatar are crucial [36], as the visitor needs to receive an almost immediate response for not breaking the immersion.

Some approaches rely on semantics and affective properties to create natural behaviors for creating an empathetic approach where the avatar can listen to the user and react to the speech if it has any mood connotations [33]. Extracting semantic tags from the text to guide the animation could be a solution by dividing them into emotion and intention tags [37]. Other approaches use recurrent neural networks (RNN) to synthesize an output video based on a learned dictionary of personal speech iconic gestures [38].

Face and lipsyncing is a solution present in avatars with bones, action unit and phoneme control [39]. These avatars are often not used in CH environments, as it would need a multidisciplinary team to develop an animation system. However, when they are included [6], they are usually not endowed with emotional capabilities.

### 2.4. Objects

When using tangible objects, the visitors can develop sensory relationships with some of them and value them as physical mementos of their interaction [2]. Some authors allow users to interact with the objects, but these have no real representations and the interaction is performed only with a virtual 3D object [18]. For example, ref. [40] investigates the augmentation of computer generated imagery with a physical replica to increase immersion and awareness; however, the interaction is by means of a laser pointer. Some museum tours have real objects. However, as these are archaeological assets, they must be placed behind security glass, so they use virtual 3D representations to allow the user to look in detail [6].

### 2.5. Gestures

Gestures can be a suitable input from the visitors to control the system, allowing their motion to be tracked and choosing what content they want to access [8]. This gesture detection is usually performed in two ways, by internal tracking of the head or hand position in an AR system [6,11], or by having a physical camera that tracks the user's body or hand gestures [11].

### 2.6. Emotional Interaction

It is well-known that our emotions influence our decision-making [41]. Therefore, it is common for an intelligent virtual tourist guide to have affective capabilities, such as emotions, mood or even a personality [42].

As shown in Table 1, few pieces of research take emotional interaction into account. As is mentioned in [11], there have been significant advancements over the last years in the digitization process with computer graphics techniques and archiving strategies. However,

this does not translate in most virtual museums, as they do not attract the public's attention and involvement: they lack stimulating activities for visitors, narratives metaphors, and emotional impact. In this work, the authors tried to create a deep emotional involvement, using dramatic lighting, storytelling, and sounds. Other studies focus on using emotional content to evoke emotions in the users, for example using virtual reality [13] to connect in a measurably emotional, participatory, interactive and social fashion, or emotional responses caused by virtual humans [7].

### *2.7. Groups Interaction*

Although group visits are common in CH environments, the current state-of-the-art in virtual museum experiences tends to focus on individual interactions due to the use of headsets or glasses [6,18]. Other interactions that use mobile apps are also single-user oriented [2,10,16].

However, visitors may vary in age with school groups formed by children that cannot follow instructions for using individual hardware such as glasses or phone apps [5]. In this case, group interaction allows them to participate more easily.

## 3. Design Methodology

A specific methodology for designing interactive tourist experiences has not been found within the related work. We propose a set of tools for designing interactive experiences for cultural heritage places in-situ, emphasizing the visitor's emotions, immersion and freedom to interact with the heritage. As a conclusion extracted from the related work, we propose a profile of experts that should participate in the design process.

### *3.1. Interdisciplinary Team*

A common issue in new media research is that it is usually constrained to one scientific field. In Section 2, we found some systems designed by an interdisciplinary team but, except for the design made by [16], the interdisciplinary team was composed of end-users and technical experts.

Interactive digital storytelling needs experts from all relevant fields to develop a new methodology that all target groups can appreciate. As in [16], we specifically address the opposing views of artistic liberty on one side, and technical constraints and historical facts on the other. As described in Section 2, designing an immersive interactive experience in CH involves several features for our research: 3D projection mapping, narration, realistic avatars, images, music, and emotions. Only a multidisciplinary team can combine these notions into an interactive application.

The main objective of the team is to achieve user immersion, the user is the central point. Our multidisciplinary team is formed by experts in designing the communication and interaction with the system to obtain user immersion in the story while taking educational, entertainment, and easy-of-use points of view. The following sections describe expertise areas required for the multidisciplinary development team.

- Computer science. To implement the proposed design with the available hardware and software, the system should run in realtime and be deployed in a computer at the museum. The computer science team must include an expert in human–computer interaction;
- Archaeologists. Interactive tangible objects should be precisely represented so that the tourist learn about the history and art of the place they are visiting;
- Historian. Content must be historically accurate; explanations and dates should be precise;
- Film arts directing. For interactive digital stories, both movies and interactive virtual environments must be taken into account. Once the story is set, the director must design the interaction in these virtual environments so users can experience the story in the best possible way;

- Screenwriter, storytelling for cultural heritage. The story is a core point of the design proposal. The experience starts when the visitors arrive at the place and lasts until they leave, so the characters involved in the story, the storytelling itself and the user interaction should make an immersive experience. Also, the historical facts provided by the historian and existing archaeological knowledge must be translated into an interactive format, so the tourists can learn in an entertaining way;
- Game directing. There should be a gamification process so visitors, both children and adults, can be entertained and are able to interact and complete tasks between short explanations;
- Visual arts, graphics design. Visuals are key for making a first impact on the user. The characters must look real and 3D animations should look natural so the uncanny valley is avoided. Other representations such as 3D environments must look accurate for the story's needs, with special attention paid to textures, 3D models and overall lighting;
- End-users: since we are designing an interactive experience for a cultural heritage place, we defined a tourist as the end user for whom the experience is designed.

*3.2. Proposed Interactive Technologies*

In this section, we propose a set of features to develop an in-situ immersive experience where the visitors are free to touch and explore the CH elements and learn about the history around them:

- In-Situ: in the state-of-the-art, several CH applications have been designed for VMs or Web; however, in this work we focus on applications designed to experience CH in the real place. The technical team must study the environmental restrictions and select a suitable hardware architecture that respects them;
- Mixed Reality (MR): there is a wide range of technologies used to introduce the user to an immersive environment. We focus on technologies that augment reality, leaving the visitor with freedom of movement. Since our aim is for the visitors to feel the experience in-situ, being free to move and interact naturally with real objects (archaeological reproductions and 3D prints) and virtual ones (virtual guide and 3D scans) is crucial. Augmentation will be achieved by projecting the digital content on the physical world employing 3D video mapping. We propose the design of a virtual projection over the real space, given the importance of physical experiences and group interaction. The projection is also supported by the use of tangible objects that are connected with a virtual replica for greater user immersion;
- Interactive digital storytelling (IDS): tourists want to learn about the history and culture of the visited place. The IDS allows the generation of the narrative through the interactive events, giving the tourist the possibility of being the director of his/her own visit;
- Avatar: the storytelling should be performed in an expressive way by a believable avatar;
- Objects: tangible objects will be placed around the space to enable the visitors to directly interact with them. This creates an emotional connection but also allows the system to detect the user's preferences and adapt the storytelling and activities;
- Gestures: to offer an interactive experience the visitor needs to interact with the system. Thanks to the IDS, we will have voice interaction but integrating a body gesture analyzer could offer more information about the visitor's preferences. We will also propose 3D printed replicas of the real objects that can be manipulated by the users as a way of interaction;
- Emotions: the interdisciplinary team must consider the visitor emotions during the design. Technology must be included in order to recognize the visitor emotions and adapt the storytelling to them;
- Groups: the in-situ visits are usually made by groups of people, so the system will need technology to manage group interaction.

*3.3. Participatory Design*

Creating experiences for a cultural heritage persona remains a challenge because no clear methodology currently exists [43]. We propose to use a participatory design method based on the User Journey Map method. Journey maps have user experience deliverables, and they have widespread usage among UX experts [44]. A User Journey is a user-centered design method mainly used in a system's definition or evaluation phases. It shows step by step the user interaction with a system describing their emotions and reactions at each of the points of contact (touchpoints) with the product.

We decided to follow this methodology for several reasons. First of all, a User Journey Map helps us to understand and communicate users' behavior as they progress through a route through interactions as they try to accomplish their goals. The tool is focused on empathy for user needs and understanding their behaviors. Journey maps also provide visualization techniques for effective communication inside multidisciplinary teams [43].

## 4. Workshop

*4.1. Preliminary Session: Persona, User Goals and Scenario*

Each User Journey Map has to reflect the itinerary of a specific person through a given scenario. Therefore, the first step was a preliminary brainstorming session for defining the persona and the task model which explains the main scenario.

This session was performed by seven persons with one expert from each discipline: computer science, archaeologists, historian, film arts director, screenwriter, game directing, visual and end-users.

The session lasted 2 h and was carried out following the approach by [44], who defined a persona using the 3i method: investigate, identify, and imagine.

The first aim of the session was to define the persona. This element will represent a group of users with similar goals. During the first part of the session, the interdisciplinary team defines Mrs. Dubois as our user target. The definition performed of Mrs. Dubois as Persona is detailed in Table 2.

**Table 2.** Mrs. Dubois definition as Persona.

| | |
|---|---|
| **Behavior** | • She often travels by car with her partner and children;<br>• She likes to visit places of cultural and historical interest;<br>• Although she is not fluent in Spanish or English, enjoys traveling to other countries, interested in learning about their history and culture;<br>• She likes new technologies and has several gadgets at home;<br>• When she is traveling, usually invests in new experiences that are entertaining and suitable for her children. |
| **Demographic** | • Female 40 years old;<br>• Gender: Female;<br>• Nationality: French;<br>• Educational Level: Professional;<br>• Occupation: Works as Engineer;<br>• Family Members: Husband and two daughters;<br>• Language: French;<br>• Civil Status: Married;<br>• Socioeconomic status: Average. |
| **Objectives** | • Discover historical and cultural sites of interest in Spain while traveling by car;<br>• Have content adapted to your language;<br>• Experiment with new technologies and find entertainment;<br>• Take away a souvenir of the experience and potentially showing it to her friends through social networks. |

The next step was was to define a task model. The task model is a story of what our persona does at each milestone of their journey. To capture the general picture of the experience, we start with a milestone before the persona's first interaction with our system [44] as is shown in Figure 1. Then, we define four milestones from the beginning to the end with user goals for each milestone. With this information the team was able to specify Mrs. Dubois's complete scenario as a long story describing her behavior from the beginning to the post-visit (Figure 2).

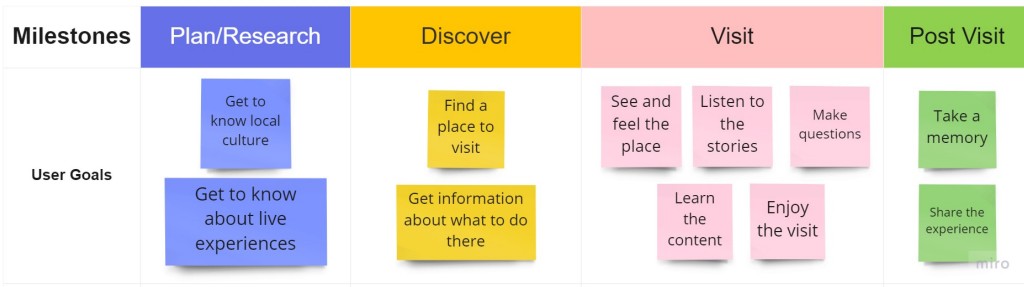

**Figure 1.** Milestones and User Goals of a visitor.

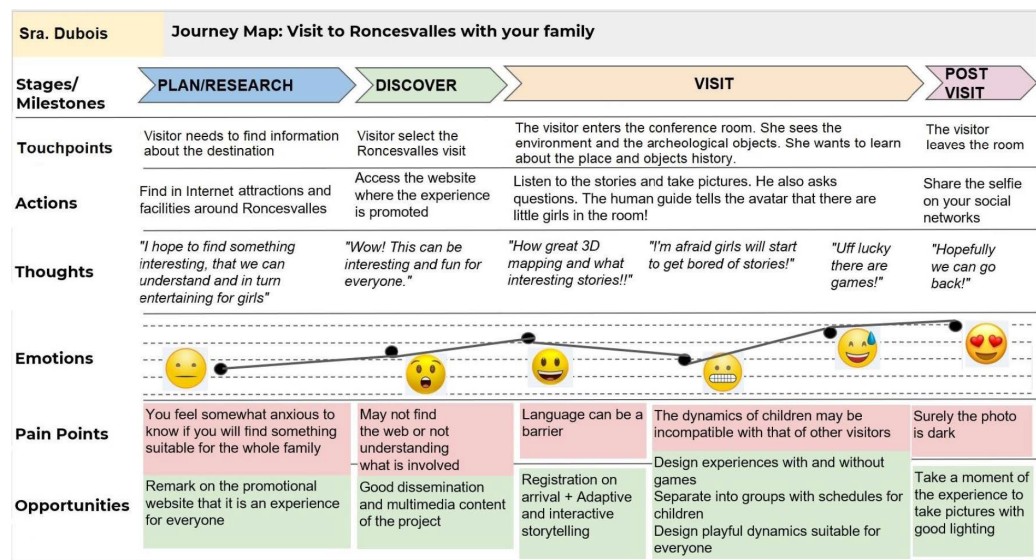

**Figure 2.** High level Mrs. Dubois's Journey Map.

The final step was to determine what information should be included in the diagram:

- Touchpoints: What part of the experience does the persona interact with?
- Actions: What does the visitor do? What information do they look for? What is their context?
- Thoughts: What is the visitor thinking?
- Emotions: What does the visitor feel during the interaction?
- Pain points: What does the visitor want to avoid? What are the obstacles?
- Opportunities: What could we improve to enhance the experience at this point?

At the end of this preliminary session, the multidisciplinary team learned how to define a User Journey Map by creating a high-level Mrs. Dubois' Journey Map.

## 4.2. Main Session: User Journey Map Design

This second session aimed at creating a more detailed view of the Visit of Mrs. Dubois' Journey Map and designing a low-level map for the interactive experience inside the Roncesvalles Place.

On this occasion, the team was made up of nine people, one history professor and two computer science professors from the Public University of Navarre. Two experts in directing and gamification came from a video game company and two archaeologists came from an archaeological company. Finally, two computer engineers also participated. Each of them took on one or more roles from the profiles defined in the interdisciplinary team. As [16] pointed out, the structure of the multidisciplinary team is inherently flexible as one person can have more than one expertise, and more than one person can share the same expertise. In this session, the specific team structure was composed of five computer scientists, two archaeologists, one historian, one film arts director, one screenwriter, one game director, one visual arts expert and five end-users.

The session lasted 3 h and was divided into three parts—30 min for contextualization, 1 h for brainstorming and 30 min for creating the User Journey Map.

The interdisciplinary team started the session with a summary of the previous sessions and the contextualization of the project. For this purpose, the team was informed about the room where the experience will be installed and the archaeological objects that will be part of the exhibition (Figure 3). Then, the computer science team introduced the interaction toolkit so that non-technical profiles could understand the technological elements that they can use during the design. The toolkit includes the following elements:

- Video Mapping 3D: Explanation of what video mapping 3D is and discussion of the opportunities that this technology can offer to achieve an immersive illusion;
- Avatar: The emotional conversational agent can chat with the tourists, selecting content according to the emotions that are to be transmitted;
- Tangible Objects. Inform that there will be archaeological reproductions in the room. When users interact with them, the system will be able to respond accordingly. We held a discussion about the type of interactions with the objects and their responses;
- Interactive Objects: The system, either through vision or through sensors, will be able to detect when a group of users is around and when they lift the objects;
- Tracking system: Groups of people could be detected; identity can be tracked by comparing the face with a photo taken at the beginning of the experience. The system will also be capable of counting some group gestures as raised hands.

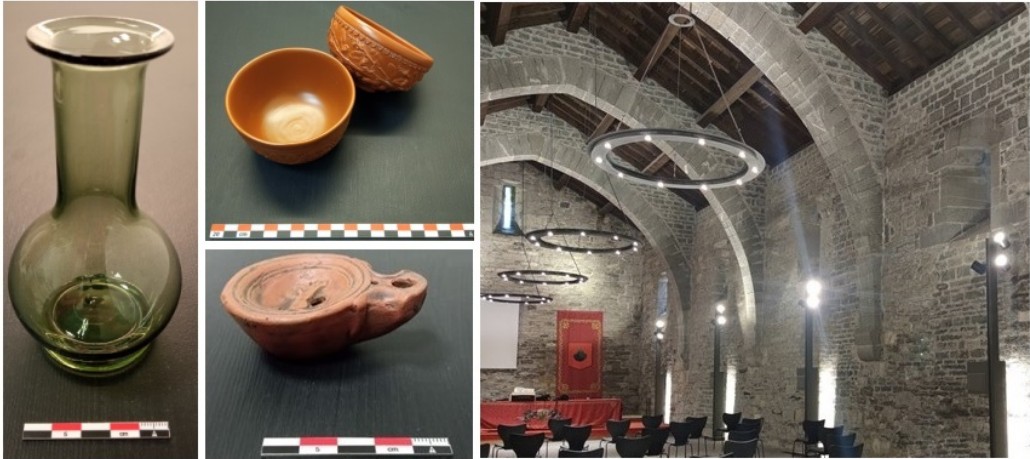

**Figure 3.** Cultural Heritage: Roncesvalles Room and Archaeological objects used as interactive objects.

Then, the interdisciplinary team held a brainstorming session following the User Journey Map Method to design an empathetic and interactive cultural heritage immersive experience. For the brainstorming session the interdisciplinary team used the materials shown in Figure 4.

Finally, the team gathered the generated ideas and designed the final experience in a User Journey Map, obtaining the result shown in Figure 5.

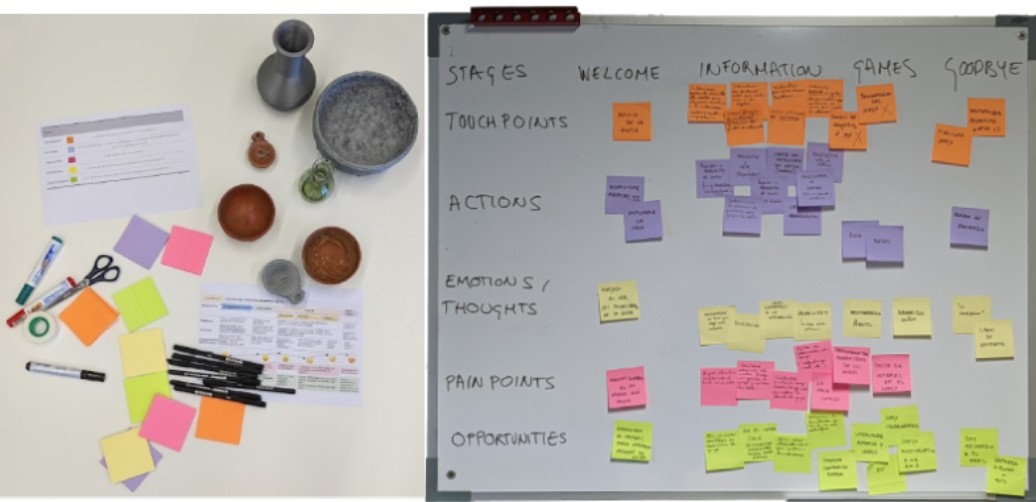

**Figure 4.** The material used during the second brainstorming session for designing an empathetic and interactive cultural heritage immersive experience.

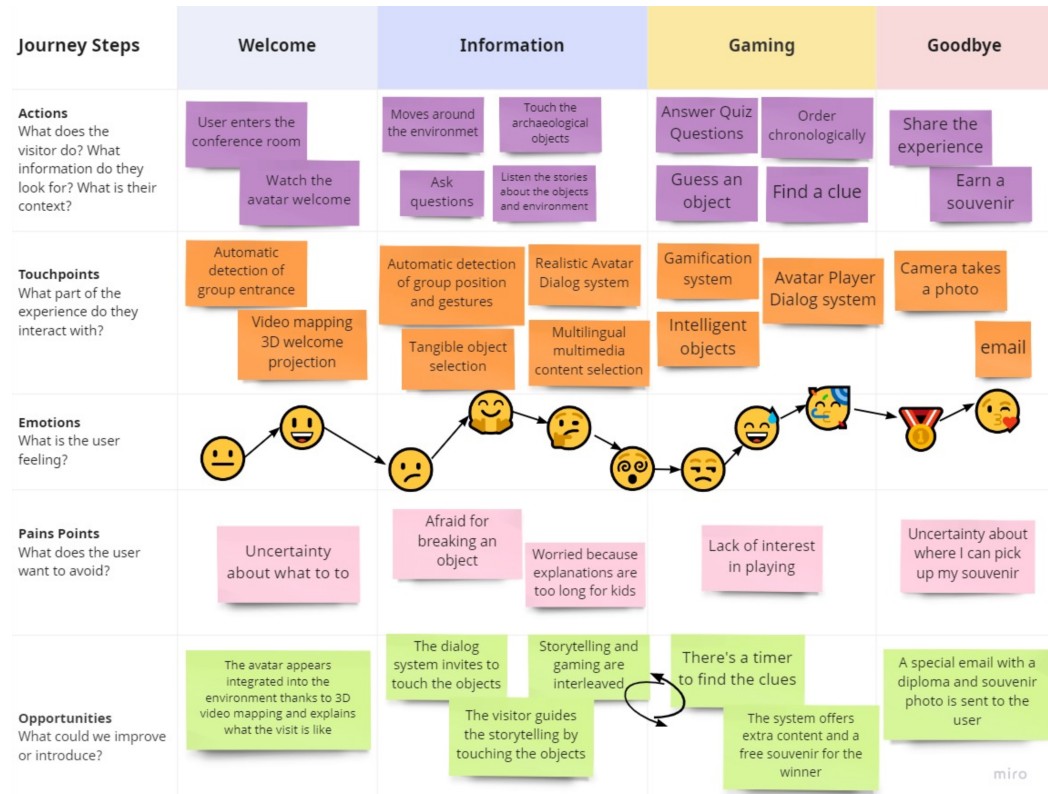

**Figure 5.** Immersive experience in Roncesvalles: User Journey Map.

*4.3. Workshop Results*

We obtained two main contributions. First, a set of guidelines for designing an immersive CH experience defined by an interdisciplinary team of experts. Second, a user-centered design system for developing an immersive CH experience in Roncesvalles. The design is shown by means of a User Journey Map and a simulation of the scene.

4.3.1. Guidelines for Designing an Immersive CH Experience

After completing the workshop, we collected a set of guidelines for designing an immersive and interactive presentation of cultural heritage provided by interdisciplinary experts.

- Experts from CH documentation: Historian and Archaeologists.

- Balance objects' appearance so they invite the users to interact with them. A realistic replica allows visitors to appreciate fine details, yet 3D printed objects encourage users to interact without fearing damaging the object;
- Real books can offer additional information. They can be used as smart objects that enhance the story. In this case, its visualization should be adapted to the group using projections and avatars;
- The information must be correctly structured, avoiding unnecessary material and giving the visitor the opportunity to choose the level of depth of the story he/she wishes to know.

- Experts from arts: graphics design and film directing.
  - For an immersive experience, 3D mapping and creating a seamless transition between the actual wall and the avatar appearance is required;
  - The avatar should be hyper-realistic to give the visitor the illusion of speaking with a guide;
  - For immersion, the scene and camera should be static for allowing spatial coherence;
  - The multimedia content displayed on the walls of the site should always be integrated by 3D mapping to avoid breaking the sense of immersion and the realism of the avatar.

- Screenwriter and storytelling for cultural heritage.
  - The story should not be linear. The visitor should choose what content they want to listen to by asking questions or selecting an object;
  - The IDS information should be structured in different levels of detail, the user being the one who decides how deep to dive into the lore that the object offers;
  - Avatar's sentences should remain short as visitors tend to get bored and disconnect when explanations are long and interaction decreases;
  - The designers should answer the following question: Who should be leading the story explanation? Is the avatar a tool for the real guide or is it the other way around? Will the human guide lead the narrative or is it driven by the interaction with the environment and its objects?
  - The storytelling should follow the segmentation principle where the material is presented in non linear segments so the user can control the narrative.

- Game directing.
  - Intertwine the story with minigames—there may be kids that will lose attention during long explanations;
  - Minigames must be of different types and complexities to attract both the adults and the children;
  - Encourage taking part with a final reward.

- Computer Science.
  - Verify that the interactions defined during the workshop are achievable with the hardware and architecture of the proposed software;
  - The arrangement of the elements inside the room (Projector, Objects, Visitors, Speakers or Cameras) should be designed following the restrictions of the CH.

- End Users.
  - Texts should be translated into different languages;
  - The visitor wants to feel comfortable during the visit and it is important to consider their emotions. For example, in the case of family tourism, the parents usually get worried if the children touch something that might break or if the content is not appropriate for children.

4.3.2. User-Centered Design Experience for Roncesvalles

For designing affective experiences for cultural heritage, it is necessary to consider the emotional journey that the tourist experiences from the beginning of the visit until the end.

We have designed an interactive experience located at Roncesvalles. Roncesvalles is a very important town within the Camino de Santiago because it is the point where the roads converge. Roncesvalles is a suitable place to make an in-situ exhibition given its geographical location and rich historical content. In this work, the archaeologists and historians designed an exhibition in the context of Roman times, since the Romans built a road that passed through the town and its history is based on legends. The exhibition was designed to be installed in the conference room at "Royal Collegiate Church of Roncesvalles's Santa María", shown in Figure 3.

As represented in the User Journey Map created in the workshop, the experience is organized into four steps—Welcome, Information, Gaming and Goodbye (Figure 5).

At the beginning, the tour group enters the room and the system automatically detects the entrance to start a wall projection with 3D video mapping. The group will notice how suddenly the wall breaks and falls apart. Behind the wall, a mine is revealed and a virtual archaeologist greets the group of visitors. The projected mine is a photogrammetry of a real mine found in the Roncesvalles area. The video is an opportunity to alleviate the visitor's uncertainty and explain how to start the visit in an immersive way.

After the welcome step, the group can move around the room to observe the environment and the replicas of archaeological objects placed in the room. The objects selected for this first prototype were a lacrimarie, a bowl and an oil lamp shown in Figure 3. They are related to the historical context selected for the visit and were found at the archaeological excavations around Roncesvalles. The system automatically detects if the tour group is around an object to trigger the avatar storytelling related to that object as is simulated in Figure 6.

When the avatar begins to tell stories related to the objects, adult visitors may think that these stories will be boring for the children and they will lose interest. However, parents are relieved when they discover that the avatar can detect children in the room and proposes activities for the children, who can play a series of games. During the design workshop the team decide to mixed storytelling with gaming in order to entertain the whole family, both adults and children. So, together with the adults, the children play mini-games that follow the gamification proposed by the multidisciplinary team, with playable activities adapted for different audiences. The mini-games designed for this experience are related to the tangible interactive objects in the room and with the storytelling explained by the avatar. For example, one mini-game is to find an object hidden in the room following the clues given during the storytelling. Another example, after listening to the avatar's narration, the tourists must order the objects chronologically. For designing the whole set of mini-games, theories such as the flow concept provided by Csikszentmihalyi [45] should also be taken into account. Therefore, the set of mini-games should present activities with enough difficulty and involving some risk.

Finally, at the goodbye step, the avatar surprises the visitors by telling them that she has taken some photos of the activities and that she can send them as a gift to their e-mails so that they can share the experience on social networks.

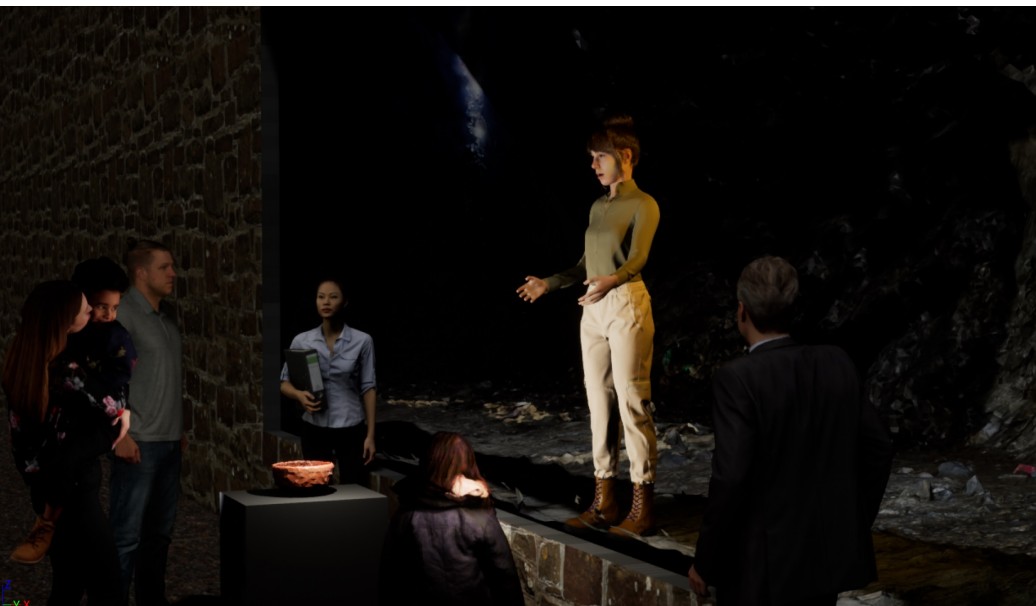

**Figure 6.** Simulation of visitors listening to the projected Avatar during the visit.

## 5. Proposed Framework

In this article, we present a design for implementing an interactive on-site experience. The proposed framework is shown in Figure 7. This design considers two inputs (visual and auditory) through which the information reaches the virtual avatar. The auditory input, in this case coming from a microphone, extracts the multimodal information perceived from the voice of the visitor, passing it to the Text module, which transcribes the audio content and the Emotional Audio Analyzer module, which detects the emotional tone of the voice.

The visual input is captured using two Kinect v2 cameras, which track the position of the visitors around the area. They also detect the interaction of the group with objects in the environment, the individual gestures and the emotion on facial expressions.

This information is sent to the interaction manager module. The information is classified, structured and sent to the dialogue manager module, which is in charge of deciding what to perform and the response it should give, as well as how it affects the avatar state.

The NLU module detects the conversational intention of the visitors and the emotional polarity of the sentence. At the same time, the personalized action based on the EBDI model establishes the beliefs related to the incoming information. Finally, it offers plans according to the situation, using the content manager and thus offering a plain text and an SSML (Speech Synthesis Markup Language) file as a response. The markup text response goes, on the one hand, to the Emotional Text-to-Speech module to generate the response audio and, on the other hand, to the the avatar module to generate the expressive animations. Finally, the system mixes the rendered avatar with the audio and then projects it onto the wall. The audio is played through the speakers.

Regarding the avatar module, there are several options for generating a believable avatar; however, in the current state-of-the-art, the Epic Games Unreal engine offers realistic avatar support called Metahumans [46]. As it is a customizable engine, it allows us to build up the proposed avatar module that will take the generated text and convert it into final face and body animations for the avatar.

The proposed avatar module takes the SSML input from the dialog manager and reads the visemes so it can generate a lipsync animation from it. Parallel to this, the lipsync plays along with the generated audio, so it can both have a voice and move the lips while reading the input.

As the dialog manager offers a conversational agent and an emotional EBDI model, it allows the export of emotional tags that will endow the avatar with emotional reactions and an empathy system to follow and react to not only the content of the user's message but its intention. These tags will generate different animations on the avatar based on emotions, intensity, and semantic features to emphasize essential terms or to be able to point out exciting objects following the story's needs.

The framework is designed to implement a sufficiently scalable prototype since user inputs are considered and the storytelling as well as the avatar animation are generated in real time from text. We note that the group sizes are limited during the regular exhibition and this limitation can also be used to ensure that the interactive experience works properly.

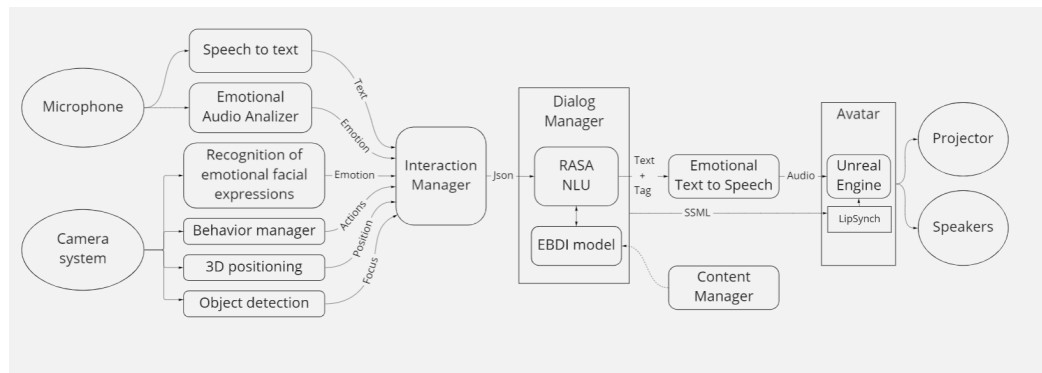

**Figure 7.** Proposed Framework for implementing an interactive on-site experience.

## 6. Conclusions

Our work designed interactive experiences for cultural heritage. We informed the system design following a participatory design approach combined with UX techniques. The whole design process was enriched by an interdisciplinary team who addressed previously suggested challenges [3] regarding the storytelling content, mixed-reality environment and the architecture of the physical space. Specifically, we contribute by summarizing the process in terms of design guidelines and recommendations for future developments.

**Author Contributions:** Conceptualization, X.O., R.G., A.O., S.M. and J.V.; methodology, A.O. and S.M.; software, X.O. and R.G.; validation, J.V., O.A. and A.M.; resources, A.O., J.V., O.A. and A.M.; writing—original draft preparation, X.O., R.G., S.M. and A.O.; writing—review and editing X.O., R.G., S.M., A.O., J.V., O.A. and A.M.; supervision, A.M.; funding acquisition, A.O. and J.V. All authors have read and agreed to the published version of the manuscript.

**Funding:** This research was funded by the R & D projects of the Government of Navarra under grant agreement No 0011-1365-2021-000063.

**Institutional Review Board Statement:** Not applicable.

**Informed Consent Statement:** Not applicable.

**Data Availability Statement:** Not applicable.

**Acknowledgments:** We thank Carlos Zuza and Nicolás Zuazúa from the archaeological company Gabinete TRAMA for their contributions for the designing of Storytelling and Tangible objects. We also thank to Javier Larumbe from Nexxyo Labs company for his contribution to arts and gamification design. Finally, we would like to thank Iñigo Mugueta of Medieval History & Historical Heritage, for his contributions to Storytelling documentation and game design.

**Conflicts of Interest:** The authors declare no conflict of interest.

## Abbreviations

The following abbreviations are used in this manuscript:

CH     Cultural Heritage
VM     Virtual Museums
VR     Virtual Reality
MR     Mixed Reality
AR     Augmented Reality
IDS    Interactive Digital Storytelling
IDT    Interdisciplinary Design Team

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
