# Peer review of "An Interdisciplinary Design of an Interactive Cultural Heritage Visit for In-Situ, Mixed Reality and Affective Experiences"

_mti, doi:10.3390/mti6070059_

Round 1

Reviewer 1 Report

In this paper the authors describe an interdisciplinary design approach for a projectiv AR cultural heritage guide. The paper is well written and I like the interdisciplinary team. To combine CH, MR and Gamification, experts from each side are necessary.

Introduction gives a good overview about the goals and the procedure.

Related work is well analised according to several elements they will use in their protoype. Here some papers that should be also considered for the analysis. The projects LifePlus and also Archeoguide are classical examples for using AR at excavations sides also in combination with virtual characters and storytelling. The third paper is a good example for using projective AR in a museum interactively:

Papagiannakis, George, et al. LIFEPLUS: revival of life in ancient Pompeii, virtual systems and multimedia. No. CONF. 2002.

Vlahakis, Vassilios, et al. "Archeoguide: an augmented reality guide for archaeological sites." IEEE Computer Graphics and Applications 22.5 (2002): 52-60.

Plecher, David A., et al. "Projective Augmented Reality in a Museum: Development and Evaluation of an Interactive Application." (2020).

The user centered design approach focuses on the users' emotions. In my opinion it should also consider the flow concept (Csikszentmihalyi). With regard to the use of minigames, possibly also the special meaning of flow in serious games.  

Prototype:
It would be interesting to get more precise information based on a concrete example. (How) can the user use the oil lamp (Figure 3) in the mine? What information is offered about this exhibit? Is the information scalable?

Minor typos and errors:

- subsubsection 3.0.1 should be changed to subsection 3.1
- citation [42] should be on page 7.
- Figure 2: destionation => destination
- page 10: Reference to the figure is broken

Reviewer 2 Report

The job presents a model example of working pipeline, clearly explained and well performed. beyond its interest concerning the application it is important for the pipeline adopted, whcich can be considered as a model for this kind of projects.

Only one remark: at the end is present a paragraph "7: future work" which is empty. Probably a typo to be edited.

Author Response

We appreciate the possitive comments from the reviewer. We have corrected the mentioned issued with the empty section by deleting it, future work was discussed on the previous section and the section future work was not needed.